# Studies of the Efficacy of Low-Dose Apatinib Monotherapy as Third-Line Treatment in Patients with Metastatic Colorectal Cancer and Apatinib’s Novel Anticancer Effect by Inhibiting Tumor-Derived Exosome Secretion

**DOI:** 10.3390/cancers14102492

**Published:** 2022-05-19

**Authors:** Lingying Zhao, Qiang Yu, Chunyi Gao, Jingzhou Xiang, Bowen Zheng, Yujie Feng, Runyang Li, Wenqing Zhang, Xiaoting Hong, Yan-yan Zhan, Li Xiao, Tianhui Hu

**Affiliations:** 1Xiamen Key Laboratory for Tumor Metastasis, Cancer Research Center, Xiamen University School of Medicine, Xiamen 361102, China; zhaolingying@stu.xmu.edu.cn (L.Z.); 24520160154575@stu.xmu.edu.cn (Q.Y.); 21620190154571@stu.smu.edu.cn (C.G.); xiangjzhou@stu.xmu.edu.cn (J.X.); zhengbowen@stu.xmu.edu.cn (B.Z.); fengyujie@stu.xmu.edu.cn (Y.F.); 24520211154655@stu.xmu.edu.cn (R.L.); wqzhang@xmu.edu.cn (W.Z.); xthong@xmu.edu.cn (X.H.); yyzhan@xmu.edu.cn (Y.-y.Z.); 2Department of Oncology, Zhongshan Hospital of Xiamen University, School of Medicine, Xiamen University, Xiamen 361004, China; 3Shenzhen Research Institute of Xiamen University, Shenzhen 518057, China

**Keywords:** low-dose, apatinib, colorectal cancer, transcriptome sequencing, exosome secretion

## Abstract

**Simple Summary:**

We assessed the efficacy and safety of low-dose apatinib monotherapy as a third-line treatment in patients with metastatic colorectal cancer. The ORR and DCR were 4.0% (2/50) and 70% (35/50), and the median PFS and OS were 4.7 months and 10.1 months, which demonstrated comparable survival outcomes, significant improvements to the patient’s quality of life, and tolerable adverse reactions. We also disclosed a novel role of apatinib’s anticancer effect, i.e., inhibiting tumor-derived exosome release. Our results indicated that apatinib treatment inhibited exosome secretion through the regulation of MVB biogenesis, transport, and fusion by regulating LAMP2, RAB11, Snap23, and VAMP2. This novel regulatory mechanism provides a new perspective for the antitumor effect of apatinib in CRC treatment.

**Abstract:**

Antiangiogenic therapy is an important treatment strategy for metastatic colorectal cancer (mCRC). We carried out a clinical study of low-dose apatinib (250 mg) monotherapy as a third-line treatment in patients with mCRC and assessed its efficacy and safety. It demonstrated that low-dose apatinib had comparable survival outcomes, significantly improved the patient quality of life, and caused tolerable adverse reactions. To further investigate the underlying mechanism of the effects of apatinib in CRC besides angiogenesis, we performed RNA-seq, and our results suggested that apatinib may have other potential antitumor mechanisms in CRC through multiple pathways, including exosomes secretion. In RKO and HCT116 cells, apatinib significantly reduced exosomes secretion by targeting multivesicular body (MVB) transport. Further studies have indicated that apatinib not only promoted the degradation of MVBs via the regulation of LAMP2 but also interfered with MVB transport by inhibiting Rab11 expression. Moreover, apatinib inhibited MVB membrane fusion by reducing SNAP23 and VAMP2 expression. In vivo, apatinib inhibited orthotopic murine colon cancer growth and metastasis and reduced the serum exosomes amount. This novel regulatory mechanism provides a new perspective for the antitumor effect of apatinib beyond angiogenesis inhibition.

## 1. Introduction

Colorectal cancer (CRC) is the second-most common adult cancer in women, the third-most common in men, and the fourth leading cause of cancer death [1,2]. A large number of patients suffer from advanced/metastatic tumors, with a 5-year survival rate of less than 20% [3]. Antiangiogenic therapy is an important treatment strategy for metastatic colorectal cancer (mCRC). In 2004, AVF2107g, the first phase 3 study, led to the approval of bevacizumab as the first targeted therapy for patients with mCRC, which is the milestone of targeting angiogenesis in mCRC [4]. Other drugs targeting angiogenesis effectors (e.g., ramucirumab and afibercept) were approved after bevacizumab failure, confirming the concept of “continuous antiangiogenic blocking” [5]. Recently, a number of new orally available multiple receptor tyrosine kinase inhibitors have been tested in late-stage clinical trials. New molecules targeting angiogenesis, such as furaquintinib and regorafenib, etc., have recently been tested in refractory mCRC in order to prevent the primary and secondary resistance of frontline antiangiogenic therapy [6].

Each antiangiogenic drug has its peculiar mechanism of action and has demonstrated efficacy and safety in pivotal and real-world studies. Apatinib is a highly selective oral inhibitor of VEGFR-2, with a putative activity as a multidrug resistance reverser as well [7]. Apatinib is currently approved in China for the treatment of advanced gastric cancer and hepatocellular cancer. Recently, many clinical studies have shown apatinib’s satisfactory efficacy in various types of cancers, including non-small cell lung cancer (NSCLC), breast cancer, esophageal cancer, hepatocellular carcinoma, sarcoma, ovarian cancer, cervical cancer, cholangiocarcinoma and colorectal cancer, etc. [8]. A pilot study of apatinib as a third-line treatment in Chinese patients with heavily treated metastatic colorectal cancer reported a disease control rate (DCR) of 77.8% and a median overall survival (OS) of 10.1 months from apatinib (425–750 mg once a day) [9]. Two prospective studies reported that apatinib (500 mg once a day) monotherapy in a third-line setting displays promising activities in patients with refractory colorectal cancer [10,11]. However, in these studies, apatinib showed severe grade 3 to 4 adverse events, and the patients were not well-tolerated and controlled. In our study, we want to provide additional clinical evidence and explore the efficacy and safety of low-dose apatinib (250 mg once a day) for third-line treatment in Chinese patients with heavily treated metastatic colorectal cancer.

The global antitumor therapeutic effect of an antiangiogenic agent is due to not only antiangiogenesis but also many other mechanisms. Multiple tumor-related kinases can be inhibited by apatinib, including VEGFR-2, platelet-derived growth factor-β (PDGF-β), rearranged during transfection (RET), etc. [7]. In anaplastic thyroid cancer cells, apatinib can induce apoptosis and autophagy via the AKT/mTOR pathway [12]. In ovarian cancer cells, apatinib can promote autophagy and apoptosis through the nrf2/ho-1 pathway [13]. The growth of small cell lung cancer (SCLC) cells can be inhibited by apatinib via the decreased expression of CD31, Ki-67,VEGF, p-AKT, p-ERK1/2, p-PI3K, and pVEGFR-2 [14]. In vitro and in vivo, gastric cancer stem cells can be inhibited by apatinib through suppressing the SHH pathway [15]. Apatinib can also inhibit glioblastomas by effectively targeting NDUFA4L2 (subunit of complex I of the mitochondrial respiratory chain) [16].

In addition to angiogenesis inhibition, some novel regulation mechanism could underline the antitumor effect of apatinib in CRC treatment. Cancer development and progression is a complex, multistep process driven by many signaling events. In particular, exosomes affect cancer progression by promoting autocrine/paracrine signaling, reprogramming stromal cells, inducing angiogenesis, and interfering with the immune system [17,18,19]. Exosomes secreted by different cells also have different functions and characteristics. In colorectal cancer cells, exosomes have been shown to promote proliferation and invasion and to make the surrounding environment more suitable for tumor growth [20,21]. Here, we have comprehensively tested the therapeutic effect of apatinib in CRC and discovered that apatinib treatment exerted exosome inhibition and antitumor efficacy for CRC cells in vivo and in vitro. Our results demonstrated that apatinib treatment could significantly alleviate CRC progression and disclosed a novel role of apatinib’s anticancer effect by inhibiting tumor-derived exosome release.

## 2. Materials and Methods

### 2.1. Patient Eligibility

The ethics committee of Zhongshan Hospital Xiamen University approved this retrospective study. All the patients or their legal guardians reviewed and signed the informed consent before treatment. All the enrolled patients were diagnosed with advanced or mCRC and progressed after at least two lines of systemic therapy. The inclusion criteria included: (1) patients were confirmed to have colorectal cancer histologically, (2) patients who were treated with apatinib in the ≥3rd line from March 2015 to August 2017, and (3) patients who had at least one measurable lesion defined and evaluated according to the Response Evaluation Criteria in Solid Tumors (RECIST) criteria (1.1).

### 2.2. Treatment

Oral apatinib was administered once daily initially. If patients had severe adverse events, the investigator could interrupt apatinib, reduce apatinib to 125 mg, or permanently discontinue the treatment.

### 2.3. Assessments of Efficacy and Safety

Progression-free survival (PFS) was set as the primary analysis endpoint; the OS, objective response rate (ORR), and DCR were secondary analysis endpoints. The time period from initiating the apatinib treatment to death or disease progression was defined as PFS. The time period from initiating apatinib treatment to the date of death due to any cause or last follow-up visit was defined as the OS. Both radiologists and oncologists assessed the tumor responses every two months or if the patients had any sign of progression. Objective tumor responses were assessed by RECIST criteria (1.1), including complete response (CR), partial response (PR), stable disease (SD), and progressive disease (PD). The addition of CR + PR + SD rates was defined as DCR. The addition of CR + PR rates was defined as ORR. According to the National Cancer Institute Common Toxicity Criteria for Adverse Events version 4.0 (CTCAE4.0), the investigators reviewed and determined the toxicities of apatinib from patients’ laboratory examination results and medical history.

### 2.4. Cell Culture and Reagents

RKO and HCT116 cell lines were obtained from the Cell Bank/Stem Cell Bank of the Chinese Academy of Science (Shanghai, China). Cells were cultured at 37 °C and 5% CO_2_ and in high-glucose PRMI with 10% fetal bovine serum (FBS). Apatinib (kindly provided by Hengrui Medicine Co. Ltd., Jiangsu, China) was dissolved with DMSO and diluted in RMPI medium for further experiments. The following antibodies were used for this study: mouse monoclonal anti-actin (#a3854), rabbit monoclonal anti-ALIX (#ab186429), mouse monoclonal anti-CD63 (#sc-5275), mouse monoclonal anti-EEA1 (#sc-137130), mouse monoclonal anti-LAMP2 (#sc-18822), rabbit polyclonal anti-Rab11a (#71-5300), rabbit polyclonal anti-Rab11b (#19742-1-AP), mouse monoclonal anti-Rab11c (#sc-65978), mouse polyclonal anti-Rab27a (#17817-1-AP), rabbit polyclonal anti-Rab27b (#13412-1-AP), rabbit polyclonal anti-Snap23 (#111 203), rabbit monoclonal anti-Tsg101 (#ab133586), and mouse monoclonal anti-VAMP2 (#104 211).

### 2.5. RNA Sequencing Assay

The RNA-sequencing assay included the cDNA library construction, library purification, and transcriptome sequencing. All experiments were finished according to the protocols of the Wuhan Huada Sequencing Company’s instructions. Each group of the RNA-sequencing assay used three samples.

### 2.6. Quantitative Real-Time PCR

Cells were treated with lycorine and harvested after 24 h. The total RNA of the cells was extracted with Trizol reagent. The FastKing RT kit was used to reverse-transcribe RNA to cDNA. QRT-PCR was performed with the GoTaq^®^ qPCR Master Mix. The glyceraldehyde-3-phosphate dehydrogenase (GAPDH) value was used as the normalized expression control. Appendix A showed all the primers.

### 2.7. Western Blot

Proteins were obtained from cell lysates with the RIPA buffer. The bicinchoninic acid (BCA) assay kit was used to determine the concentration of the protein. The total lysates were subjected with sodium dodecyl sulfate (SDS) loading buffer at 100 °C and then exposed to sodium dodecyl sulfate polyacrylamide gel electrophoresis (SDS-PAGE). Polyvinylidene difluoride (PVDF) membranes were transferred with the protein and blocked with 5% dried skimmed milk. Specific primary antibodies were incubated with PVDF membranes overnight. Then, HRP-conjugated secondary antibody was added, and the expression of the protein was examined with the ECL plus reagents.

### 2.8. Go and Kegg Enrichment Analysis

GO and KEGG enrichment analyses revealed the functional roles of differentially expressed genes (DEGs). The GO database was firstly used to analyze the functional enrichment of DEGs enriched in terms of the cellular component, molecular function, and biological process. The enrichment of the DEGs was determined by the KEGG pathway database. The threshold was 0.05 to test the hypergeometric distribution of the default enrichment results.

### 2.9. Isolation of Exosome

We prepared exosomes from the culture supernatant from CRC cells after 48-h culture. To remove debris, we centrifuged the culture supernatant at 2000× *g* for 10 min and then 10,000× *g* for 30 min at 4 °C. The supernatant was then centrifuged at 100,000× *g* for 70 min at 4 °C. The pellet was washed with phosphate-buffered saline (PBS). Again, the supernatant was centrifuged at 100,000× *g* for 70 min at 4 °C. Then, we collected the exosomes and resuspended them in PBS [22].

### 2.10. Nanoparticle Tracking Analysis

The NanoSight NS 300 system (NanoSight Technology, Malvern, UK) was used to track and analyze the number and size of the exosomes. Dulbecco’s PBS (DPBS) without any nanoparticles was used to dilute the samples 150–3000 times to a concentration of 1–20 × 10^8^ particles/mL for analysis. We measured each sample three times and recorded each visible particle. The Stokes–Einstein equation was used to analyze the exosome numbers and size distribution.

### 2.11. Immunofluorescence Assays

Four percent paraformaldehyde was used to fix the cells at 25 °C for 25 min. Then, the indicated primary antibodies (1:100) were used to stain the cells overnight at 4 °C. Secondary antibodies were incubated with the cells at 37 °C for 1 h. 2-(4-amidinophenyl)-1H-indole-6-carboxamidine (DAPI) (Beyotime) was used to stain the nuclei at room temperature for 3 min. A Nikon A1r confocal microscope was used to capture the immunofluorescence.

### 2.12. Transmission Electron Microscopy

Five percent glutaraldehyde in 0.1 M phosphate buffer was used to fixed the cells at 4 °C (protected from light). Slices of about 70 nm in thickness were generated by the fixed cells subjected to ultra-cryomicrotomy. We placed the isolated exosomes in heavy suspension droplets on the special copper mesh of the electron microscope. Then, 20 μL of 2% phosphotungstic acid was subjected to negative staining for 10 min. An H-7650 electron microscope was used to analyze all the samples at 100 KV.

### 2.13. Animal Experiments

All the protocols of the animal experiments were approved by the Animal Care and Use Committee of Xiamen University. A certain amount (100 μL) of RKO-luc cell suspension with a concentration of 2.5 × 10^6^/mL was injected subcutaneously into the right dorsal flank of BALB/c nude mice (5-week-old females). When the subcutaneous tumor tissue was macroscopic, we dissected the subcutaneous tumor tissues and embedded them into the mesentery of BALB/c nude mice (5-week-old females). We divided the mice into two groups: the control group and the apatinib group (5 mice/ group). The control group mice were administered a daily oral gavage with 50 mg/kg PBS; the apatinib group mice were administered a daily oral gavage with 50 mg/kg apatinib. All mice were sacrificed at 40 days after RKO-luc cell inoculation, and the tumors were weighed and fixed for immunohistochemistry staining. Mice were anesthetized and intraperitoneally injected with 1.5 mg D-Luciferin of 100 μL PBS for the living imaging assay. The IVIS Lumina II apparatus (PerkinElmer, Waltham, MA, USA) was used to analyze the bioluminescence. The mice sera were collected, diluted with PBS, and centrifuged at 10,000× *g* for 30 min at 4 °C. A 0.22-μm filter was used to filter the resultant supernatant fluid; then, the supernatant was centrifuged at 200,000× *g* for 2 h at 4 °C. The pellets were washed with PBS and then centrifuged at 200,000× *g* for 1 h at 4 °C to purify the exosome.

### 2.14. Statistical Analysis

SPSS (Version 24.0, Chicago, IL, USA) and GraphPad Prism (Version 8.0, San Diego, CA, USA) were used to analyze the data. The results were presented as the mean ± standard deviation (SD). One-way ANOVA was used to determine the differences among the groups following Tukey’s post-test. OS and PFS were estimated by the Kaplan–Meier method with a log-rank analysis.

## 3. Results

### 3.1. Low-Dose Apatinib Monotherapy as Third-Line Treatment in Patients with Metastatic Colorectal Cancer

Although fruquintinib, regorafenib, and TAS-102 are recommended now for patients with mCRC after the failure of standard treatments in China, they were not approved during our low-dose apatinib study period. This was a retrospective real-world study, and 50 patients were enrolled. All patients had colorectal cancer with liver, lung, lymph node, peritoneum or other metastases and had previously been treated with standard first- and second-line therapies. The main clinicopathological characteristics of the 50 patients are shown in Table 1.

All 50 patients took 250 mg apatinib orally once daily and were evaluated by image examination. Two (4.0%) patients achieved PR, thirty-three (66.0%) patients had SD, and fifteen (30.0%) patients were reported with PD after the treatment, respectively, with an ORR of 4.0% and DCR of 70.0% (Figure 1A). The median PFS and median OS with the low-dose apatinib treatment were 4.7 (95% confidence interval (CI): 3.1–7.9) and 10.1 months (95% CI: 8.6–12.8), respectively (Figure 1B,C). The subgroup analysis showed that there was no significant difference in the median PFS (4.45 months vs. 4.75 months, *p* = 0.078) and median OS (10.6 months vs. 9.95 months, *p* = 0.302) with patients who were previously treated with and without bevacizumab (first and second standard antiangiogenic therapies) (Figure 1D,E), indicating that patients could benefit from apatinib treatment regardless of whether the patient had previously received bevacizumab treatment or not.

The adverse event data of low-dose apatinib are summarized in Table 2, including hypertension, hand–foot syndrome, proteinuria, dysphonia, mucositis oral, decreased appetite, fatigue, diarrhea, neutropenia, anemia, and platelet count decreased. The grade ≥3 severity adverse events most frequently reported were hypertension in 4 out of 50 patients (8%), hand–foot skin reaction in 3 patients (6%), and proteinuria in 1 patient (2%) (Table 2). Sixteen percent of all the patients developed grade 3 adverse events. Grade 4 toxicity was not observed in all the patients. Toxicities of low-dose apatinib treatment were generally well-tolerated and controllable via symptomatic treatment. No patient discontinued the apatinib treatment due to adverse reactions, and none of the patients died of drug-related causes.

### 3.2. Global Analysis of Gene Expression and Fold Change Analysis for the Anti-CRC Activity of Apatinib

The comparable effects of low-dose apatinib in treating colorectal cancers might suggest some novel mechanisms involved besides angiogenesis. Our other studies also indicated that low-dose apatinib enhanced the efficacy of anti-PD-1 treatment in colon cancers (unpublished data), which further suggests apatinib’s potential novel roles. Therefore, we next carried out an RNA-seq study to explore the possible novel roles of apatinib in colon cancer. RKO cells with apatinib treatment were analyzed by RNA-seq, and we analyzed the gene expression profiles of the 20 μM apatinib-treated group and untreated control group. Seven hundred and fifty-five differentially expressed genes (DEG) were found, and we selected the biologically meaningful genes from differentially expressed gene (DEG) following apatinib treatment according to the criteria of log2 fold changes (FCs) > 2 and a false detection rate (FDR) of less than 0.05. Among the 755 DEGs, apatinib upregulated 454 genes and downregulated 301 genes in the RKO cell (Figure 2A). Figure 2B showed the heatmaps of the distributions of the DEGs. Figure 2C,D presented the top 50 downregulated and top 50 upregulated genes. To confirm the results of the RNA-seq, we performed real-time PCR assays. Consistent with RNA-seq data, the mRNA expressions of *AP3S2* (adaptor related protein complex 3 subunit sigma 2, *p* = 0.000), *YPEL4* (yippee-like 4, *p* = 0.007), and *SULT1A3* (sulfotransferase family 1A member 3, *p* = 0.009) were upregulated significantly in RKO cells (Figure 2E). The mRNA expressions of *TCEAL7* (transcription elongation factor A-like 7, *p* = 0.000), *TRIM59* (tripartite motif containing 59, *p* = 0.000), and *CCR4* (C–C motif chemokine receptor 4, *p* = 0.000) were reduced significantly in RKO cells after apatinib treatment, compared to untreated cells (Figure 2F).

### 3.3. GO and KEGG Analyses of Degs Mediated by Apatinib

To characterize the potential functional pathways altered by apatinib, GO and KEGG pathway enrichment analysis were performed. The GO analysis identified 20 GO terms that were significantly enriched (*q*-value < 0.05, Figure 3A), including terms related to the molecular function, biological process, and cellular components. The molecular function analysis showed that the top 20 terms of DEGs were related to nucleosome, endoplasmic reticulum–Golgi intermediate, transcription factor TFIIIB complex, Golgi cis cisterna, Golgi membrane, exocyst, ubiquitin ligase complex, protein phosphatase type 1 and 2A complex, exosome, tRNA–intron endonuclease complex, nuclear chromosome, mitochondrial proton-transporting ATP synthase, storage vacuole, transcription factor TFIIH core complex, proteasome complex, kinetochore, etc. There were 20 pathways with a *q*-value < 0.05 figured out by the KEGG pathway analysis of the DEGs shown in Figure 3B. Among them, the ten most significantly enriched pathways were viral carcinogenesis, steroid biosynthesis, biosynthesis of antibiotics, PI3K-Akt-signaling pathway, human papillomavirus infection, alcoholism, biosynthesis of secondary metabolites, metabolic pathways, synthesis and degradation of ketone bodies, and mTOR signaling pathway.

### 3.4. Apatinib Inhibits Secretion of Exosomes in CRC Cells

The results of the DEGs analysis suggested that exosomes are related to the anti-CRC activity of apatinib. To study whether apatinib inhibit exosome secretion, RKO and HCT116 cells were treated with 20 μM apatinib for 48 h. Then, exosomes from the supernatant of apatinib-treated RKO and HCT116 cells were isolated, and exosomes from wild-type RKO and HCT116 cells were used as the controls. The TEM analysis showed that these purified vesicles samples were oval and globular (Figure 4A). The total exosomal protein was significantly decreased in apatinib-treated RKO cells (average 2.40 ± 0.06 vs. 0.77 ± 0.09 mg/mL, *p* = 0.0001, apatinib vs. control) and HCT116 cells (average 1.87 ± 0.09 vs. 0.53 ± 0.08 mg/mL, *p* = 0.0004, apatinib vs. control) compared with the control cells (Figure 4B). Nanoparticle tracking analyses (NTA) revealed that apatinib-treated Exo-RKO and Exo-HCT116 had a more significant decrease in the total number than the untreated controls: Exo-RKO (average 6.90 ± 0.59 vs. 3.20 ± 0.42 × 10^8^ particles/10^7^ cells, *p* = 0.0068, apatinib vs. control) and Exo-HCT116 (average 6.27 ± 0.37 vs. 3.2 ± 0.38 × 10^8^ particles/10^7^ cells, *p* = 0.0044, apatinib vs. control) (Figure 4C). The NTA revealed that apatinib-treated Exo-RKO and Exo-HCT116 have a slightly bigger size than the untreated controls (Exo-RKO: 70.0 ± 2.9 vs. 108.3 ± 4.4 nm, *p* = 0.0019, apatinib vs. control; Exo-HCT116: 64.3 ± 4.7 vs. 98.3 ± 4.3 nm, *p* = 0.0062, apatinib vs. control) (Figure 4D,E). As the exosomes were isolated from equivalent numbers of cells, the intensity of the exosomal markers reflected the ability of cells to secrete exosomes. The exosome markers (Alix, CD63, and Tsg101) were highly abundant in the exosomes from the RKO and HCT116 cells. However, apatinib-treated cells led to a significant decrease of Alix, Tsg101, and CD63 in Exo-RKO and Exo-HCT116, suggesting that the differences in the secretion levels were due to the impaired secretion of exosomes (Figure 4F and Appendix A). These data indicate that apatinib is able to decrease the secretion of exosomes from CRC cells and that the apatinib-treated cells can secrete slightly larger exosomes.

### 3.5. Apatinib Inhibits Exosome Secretion of CRC through the Regulation of MVB Biogenesis and Targeted Transport

MVBs contained intraluminal vesicles (ILVs). When MVBs fused with the plasma membrane, ILVs were released to form exosomes. To study the molecular mechanism for the impaired exosome secretion of apatinib in CRC cells, we performed an electron microscopic analysis to investigate the number and morphology of ILVs and MVBs. We investigated whether apatinib changed the number and distribution of MVBs in CRC cells. The TEM experiments showed that the number of MVBs per cell dramatically decreased compared with the control cells: MVB per cell in RKO cells (1.667 ± 0.333 vs. 4.0 ± 0.577 nm, *p* = 0.0249, apatinib vs. control) and in HCT116 (1.333 ± 0.333 vs. 3.333 ± 0.333 nm, *p* = 0.0132, apatinib vs. control) (Figure 5A,B). However, the number of ILVs per MVB did not show any apparent alterations: ILVs per MVB in RKO cells (17.33 ± 0.881 vs. 18.00 ± 1.155 nm, *p* = 0.071, apatinib vs. control) and in HCT116 (16.00 ± 0.577 vs. 15.00 ± 0.647 nm, *p* = 0.067, apatinib vs. control) (Figure 5C). This data indicated that apatinib treatment may inhibit exosome biogenesis through regulating MVB but not through mediating ILV formation. CD63, a tetraspanin protein that marks the intraluminal vesicles (ILVs) of MVBs, was increased upon apatinib treatment (Figure 5D). Meanwhile, the immunofluorescence analysis demonstrated the perinuclear clustering of CD63^+^ MVB in RKO cells (*p* = 0.0036, apatinib vs. control) and in HCT116 cells (16.00 ± 0.577 vs. 9.00 ± 0.577 nm, *p* = 0.072, apatinib vs. control) (Figure 5E–G). These results suggest that apatinib treatment in CRC cells significantly inhibits the targeted transport of MVB, which, in turn, further inhibits exosome secretion.

### 3.6. Apatinib Promotes Degradation of MVBs via Regulating LAMP2

The increased number of MVBs could be due to either a massive increase in the biogenesis of MVBs or the inhibited transport and degradation of MVBs. To investigate the basic molecular mechanisms of how apatinib regulates MVB and exosome release, firstly, we compared the expression of the early endosome marker EEA1 in apatinib treatment cells and control cells. Western blot and RT-PCR analyses did not show any apparent alteration of EEA1 in both RKO and HCT116 cells (Figure 6A–D and Appendix A), which may rule out the influence of apatinib in the formation of MVB. Secondly, MVBs can either be directed to lysosomes, where their content is degraded, or be transported to the plasma membrane for exosome release. After apatinib treatment, Western blot analyses showed a significant increase in the protein levels of LAMP2 (a marker of lysosomes) both in RKO (*p* = 0.0027, apatinib vs. control) and HCT116 cells (*p* = 0.0161, apatinib vs. control). The RT-PCR analyses also showed that a significant increase in the mRNA levels of *LAMP2* in both RKO (*p* < 0.0001, apatinib vs. control) and HCT116 cells (*p* = 0.0032, apatinib vs. control). The immunofluorescence analysis in the RKO and HCT116 cells demonstrated an apparent increased expression of LAMP2 and the perinuclear clustering of LAMP2 after apatinib treatment (Figure 7A,B). Taken together, these data suggested that apatinib promoted the lysosomal degradation of the MVBs.

### 3.7. Apatinib Regulates MVB Transport through the Rab11-Dependent Trafficking Pathway

Previous studies showed that several Rab GTPases play a vital role in regulating MVB transport and influence the docking process. The first Rab GTPase shown to be involved in exosome secretion was Rab11. Figure 6 showed that apatinib decreased the levels of Rab11a, 11b, and 11c in both the mRNA and protein levels in HCT116 and RKO cells. The effects of Rab27a and Rab27b on exosome release have been reported in several studies using different cell lines. However, the Western blot and RT-PCR analyses did not show any apparent alterations of *Rab27a* in both the RKO and HCT116 cells after apatinib treatment (Figure 6 and Appendix A).

### 3.8. Apatinib Inhibited MVBs Membrane Tethering by Targeting Expression of Snap23 and VAMP2

The final and key step of exosome secretion is that MVBs fused with the plasma membrane and then released ILVs as exosomes. The final step of exosome secretion is mediated by key Soluble N-ethylmaleimide-sensitive fusion factor attachment protein receptor (SNARE) proteins. The SNARE complex comprises v-SNAREs (vesicle-associated membrane protein (VAMP)2, VAMP3, VAMP7, and VAMP8) on membranes of the MVBs and t-SNAREs (synaptosome-associated protein 23 (Snap)23) on the cell membrane. They form a stable ternary complex that mediates exosome secretion.

After apatinib treatment, Western blot analyses showed a significant decrease in the protein levels of Snap23 both in RKO (*p* = 0.0056, apatinib vs. control) and HCT116 cells (*p* = 0.004, apatinib vs. control). RT-PCR analyses also showed a significant decrease in the mRNA levels of Snap23 in both RKO (*p* = 0.0007, apatinib vs. control) and HCT116 cells (*p* = 0.0024, apatinib vs. Control) (Figure 6). Immunofluorescence in the RKO and HCT116 cells investigated the colocalization of Snap23 with CD63, the marker of MVB. Distinct colocalization signals between Snap23 and CD63 were significantly weakened after apatinib treatment (Figure 7C).

We also wanted to investigate if the other core family members of the SNAREs and VAMPs were involved in the MVB trafficking pathway of apatinib-functioned exosome release in CRC. Firstly, the RT-PCR analyses did not show any apparent alteration of sytaxin4 in both RKO and HCT116 cells after apatinib treatment. The sytaxin4 protein level (*p* = 0.013, apatinib vs. control) increased in apatinib-treated HCT116 cells, with no apparent alteration in the apatinib-treated RKO cells. Western blot analyses showed a significant decrease in the protein levels of VAMP2 in both RKO (*p* = 0.0018, apatinib vs. control) and HCT116 cells (*p* = 0.0004, apatinib vs. control). The RT-PCR analyses also showed a significant decrease in the mRNA levels of *VAMP2* in both RKO (*p* = 0.0001, apatinib vs. control) and HCT116 cells (*p* = 0.0004, apatinib vs. control). The RT-PCR analyses did not show any apparent alterations of *sytaxin4* in both RKO and HCT116 cells after apatinib treatment. The sytaxin4 protein level (*p* = 0.013, apatinib vs. control) increased in apatinib-treated HCT116 cells, with no apparent alteration in the apatinib-treated RKO cells (Figure 6). Taken together, these data suggested that apatinib inhibited the expression and function of Snap23 and VAMP2 during the apatinib-functioned exosome release.

### 3.9. Treatment of Apatinib Exerted Exosome Inhibition and Antitumor Efficacy for CRC Cells In Vivo

To study the therapeutic efficacy of apatinib, an orthotopic murine colon cancer model was established. The tumors were detected in the colon and other organs of the mice by in vivo bioluminescent images. As shown in Figure 8A, the fluorescence intensity increased in the control group, along with a larger tumor volume (*p* = 0.0003, Figure 8B) and higher tumor weight (*p* = 0.0121, Figure 8C). Meanwhile, the fluorescence intensity of primary and metastatic tumors, including the liver, lymph nodes, and peritoneum metastasis, also significantly decreased in the apatinib treatment group (*p* = 0.04, Figure 8D). Moreover, the concentration of serum exosomes (*p* = 0.0007, Figure 8E) and total exosomal protein (*p* = 0.005, Figure 8F) were significantly higher in the control mice than apatinib-treated mice, indicating that apatinib inhibited the secretion of exosomes.

## 4. Discussion

Angiogenesis plays a key role in colorectal cancer development, and antiangiogenic therapies have improved the colorectal cancer prognosis within the past 15 years. In China, fruquintinib and regorafinib are approved as third-line standard treatments for advanced colorectal cancer. However, other antiangiogenic TKI drugs also show good efficacy, which can provide more choices for the systematic treatment of colorectal cancer patients after drug resistance. Apatinib monotherapy has shown therapeutic efficacy in the posterior line of colorectal cancer in previous studies. However, the incidence of treatment-related adverse events (TRAE), especially those TRAE ≥ G3, is high, which reduces the quality of life and treatment compliance of patients. In this retrospective study, we assessed the efficacy and safety of low-dose apatinib monotherapy as a third-line treatment in patients with metastatic colorectal cancer. It was found that the ORR and DCR of patients after low-dose apatinib monotherapy treatment were 4.0% (2/50) and 70% (35/50), respectively. According to the follow-up results, the median mPFS and mOS were 4.7 months and 10.1 months, respectively, which demonstrated comparable survival outcomes.

Small-cellular multi-kinase inhibitors have been investigated in multiple randomized controlled trials (RCTs) for the role of ≥third-line therapies in mCRC. Nowadays in China, regorafenib and fruquintinib were approved as the standard third line treatment for mCRC. CORRECT and CONCOUR studies showed that the PFS and OS improved by using regorafenib compared to the placebo (median PFS and OS, 3.2 vs. 1.7 and 8.8 vs. 6.3 months, respectively) or BSC (median PFS and OS, 1.9 vs. 1.7 and 6.4 vs. 5.0 months, respectively) in third-line therapies in mCRC [23]. In China, fruquintinib was approved as an standard third-line treatment based on superior survival compared to the placebo, with a PFS (3.7 months) and OS (9.3 months) [24].

Although the ORR of low-dose apatinib, 250 mg was lower than that of 500 mg apatinib, the survival results and DCR were consistent with those of recently published studies of 500 mg apatinib in a similar population of patients with mCRC conducted by MM et al. (ORR: 8.51%, DCR 72.34%, PFS: 3.7 months, and OS:7.3 months) [11]; Wang et al. (ORR: 8.3%, DCR 68.8%, PFS: 4.8 months, and OS:9.1 months) [25]; and Liang et al. (ORR: 11.1%, DCR 77.8%, PFS:4.8 months, and OS: 10.1 months) [9]. Adverse events were the most common reason for dose modification, including dose reduction and interruption [26,27]. The adverse event data of low-dose apatinib in our study, including hypertension, hand–foot syndrome, proteinuria, dysphonia, mucositis oral, decreased appetite, fatigue, diarrhea, neutropenia, anemia, and platelet count, decreased. No patient discontinued the apatinib treatment due to adverse reactions, and none of the patients died of drug-related causes. Sixteen percent of all the patients developed grade 3 adverse events. Grade 4 toxicity was not observed in all patients. The incidence and grade of adverse reactions of low-dose apatinib were significantly lower than those of 500 mg apatinib [28,29,30]. In conclusion, low-dose apatinib treatment is effective in the third-line treatment of advanced colorectal cancer, which significantly improves the patient quality of life and causes tolerable adverse reactions. However, the clinical study of this paper is a retrospective study. The retrospective case series study has no premade scientific research design, so there is a lack of corresponding data in many aspects, such as an analysis, and has no control group. In the future, we hope to conduct prospective research and parallel controlled research.

Apatinib is a highly selective tyrosine kinase inhibitor to VEGFR-2, which exerts a promising antitumor effect in various tumors. Previous studies have shown that blocking ER stress-involved autophagy could significantly enhance apatinib-induced CRC cell line apoptosis both in vitro and in vivo [31], indicating that apatinib might have other anticancer mechanisms besides angiogenesis. Therefore, we performed a RNA-seq, and the transcriptomic analysis suggested that apatinib has other potential antitumor mechanisms in CRC through multiple pathways, including the regulation of exosomes secretion.

Exosomes are nanovesicle-shaped particles secreted by various cells, including cancer cells. Recently, the interest in exosomes among cancer researchers has grown enormously for their many potential roles in promoting cancer development and progression [32]. In our study, apatinib inhibited the secretion of tumor exosomes from CRC cells in vitro. Moreover, the treatment of apatinib exerted exosome inhibition and antitumor efficacy for CRC cells in vivo. Exosomes originate from the endosomal system as ILVs and are released via the fusion of MVBs with the cell membrane [33]. Our study showed that apatinib significantly inhibited the targeted transport of MVB in CRC cells, which then further inhibited exosome secretion.

It is interesting that apatinib regulates many proteins related to MVB transport, such as the downregulation of Rab11s, VAMP2, and Snap23, which are closely related with MVB transport and membrane fusion, and up-regulation of LAMP2, which is related to MVB lysosomal degradation. Taken together, apatinib inhibits MVB transporting and accelerates MVB degradation, which, in turn, attenuates exosomes secretion. The mechanisms involved in apatinib’s effects on those protein regulations need to be further studied, especially the downstream pathways of VEGFR-2.

## 5. Conclusions

In conclusion, our retrospective study firstly showed that low-dose apatinib (250 mg once daily) monotherapy treatment is effective in the third-line treatment of advanced colorectal cancer, significantly improves the patient’s quality of life, and causes tolerable adverse reactions. Notably, our study first confirmed that apatinib treatment inhibited exosome secretion through the regulation of MVB biogenesis; transport and fusion by regulating LAMP2; RAB11 (RAB11a, RAB11b, and RAB11c); Snap23; and VAMP2. This novel regulatory mechanism provides a new perspective for the antitumor effect of apatinib in CRC treatment.

## Figures and Tables

**Figure 1 cancers-14-02492-f001:**
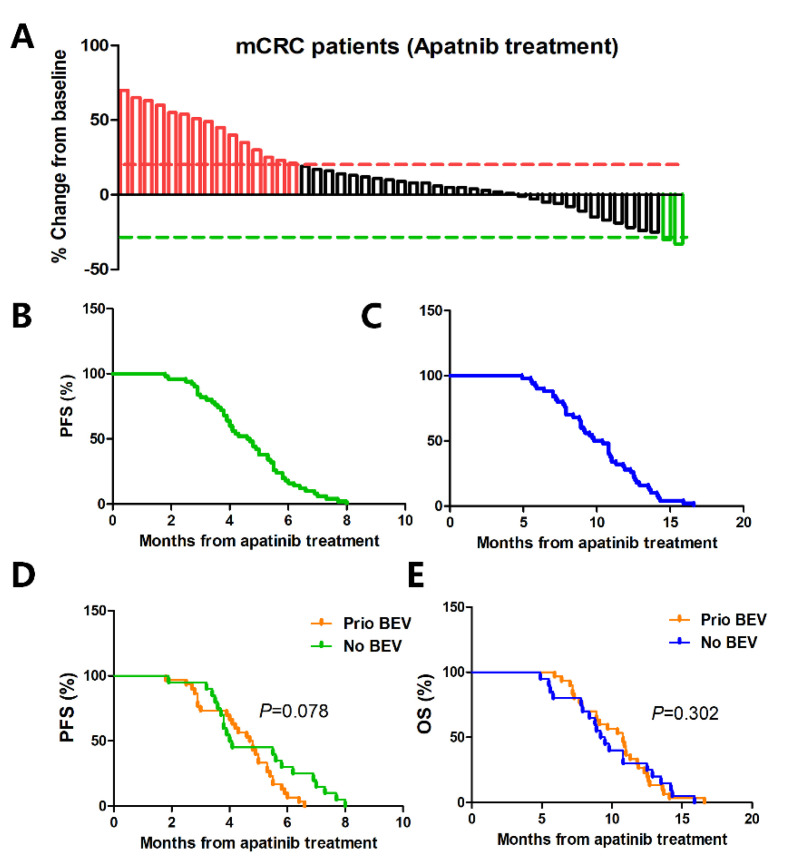
Antitumor activity of low−dose apatinib monotherapy as the third−line treatment in patients with metastatic colorectal cancer. (**A**) Best percentage changed from the baseline in measurable tumor lesions shown by a waterfall plot. (**B**) Kaplan−Meier survival curve of the PFS of the patients from low−dose apatinib treatment. (**C**) Kaplan−Meier survival curve of OS of the patients from the low−dose apatinib treatment. (**D**) PFS and (**E**) OS in patients who were previously treated with or without bevacizumab. mCRC = metastatic colorectal cancer, OS = overall survival, and PFS = progression−free survival.

**Figure 2 cancers-14-02492-f002:**
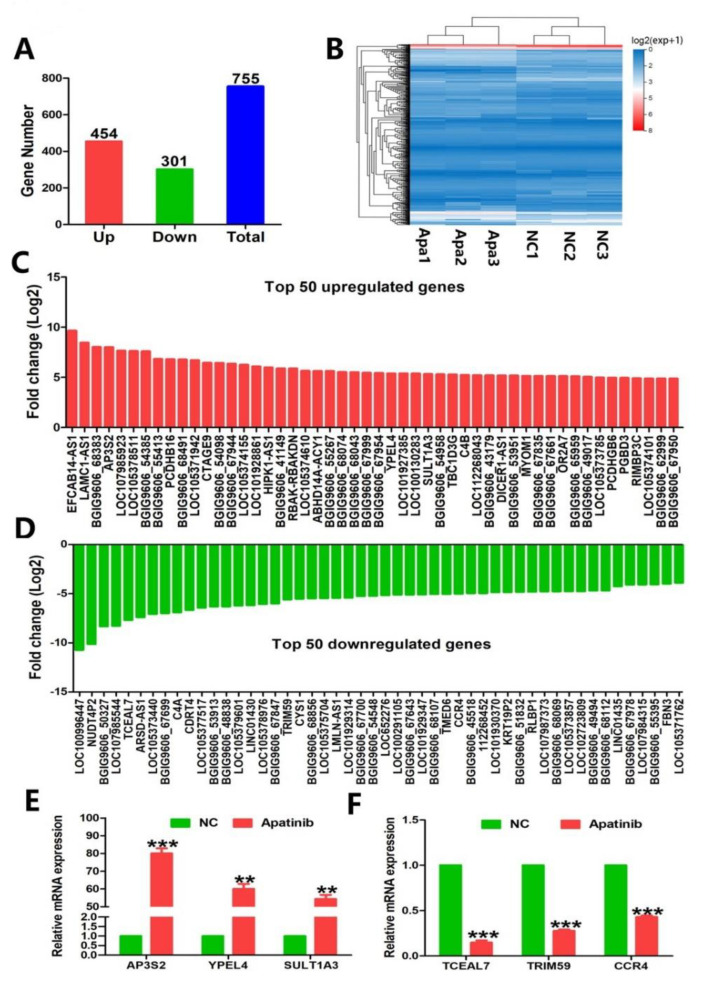
Transcriptional profiles of apatinib−treated RKO cells. (**A**) The number of genes upregulated and downregulated by apatinib. (**B**) Heatmap of differentially expressed genes between the normal group and CRC group. (**C**,**D**) Bar graphs showing the top 50 upregulated genes and top 50 downregulated genes in apatinib-treated RKO cells. (**E**) RKO cells were treated with apatinib at 20 μM for 48 h, and the mRNA expression of upregulated genes *AP3S2, YPEL4*, and *SULT1A3* was examined by real−time PCR assays. (**F**) RKO cells were treated with apatinib at 20 μM for 48 h, and the mRNA expression of downregulated genes *TCEAL7, TRIM59*, and *CCR4* was examined by real-time PCR assays. ** *p* < 0.01, and *** *p* < 0.001.

**Figure 3 cancers-14-02492-f003:**
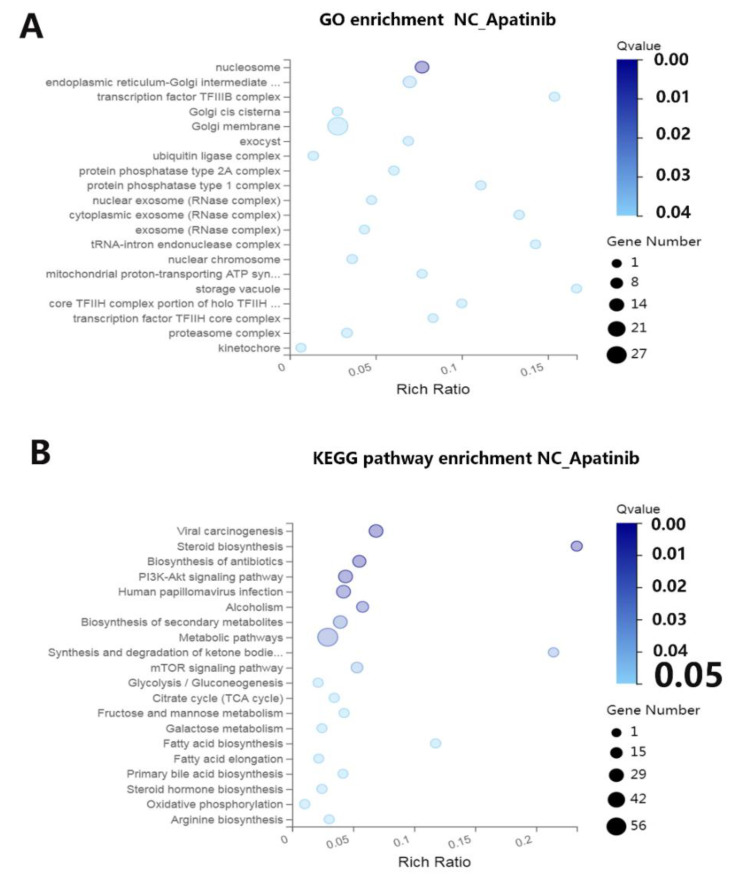
Functional analysis of the differentially expressed genes. (**A**) Gene ontology (GO) enrichment and (**B**) Kyoto Encyclopedia of Genes and Genomes (KEGG) pathway enrichment analysis of differentially expressed genes between the control group and apatinib group.

**Figure 4 cancers-14-02492-f004:**
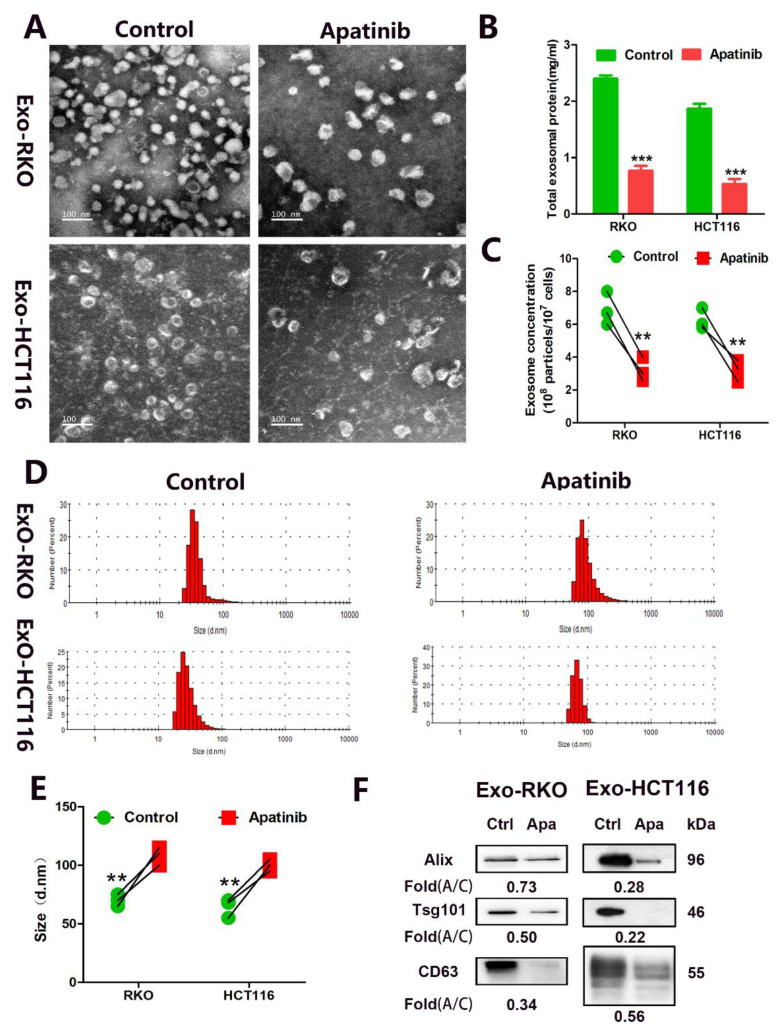
Apatinib inhibited exosome secretion in CRC cells in vitro. (**A**) Isolated exosomes from RKO and HCT116 cells were assessed by transmission electron microscopy. (**B**) Quantification of the exosomal protein levels obtained from control and apatinib treatment cells. (**C**–**E**) NTA and NTA analysis of the effect of apatinib on exosome release in RKO and HCT116 cells. (**F**) Western blot analysis of exosome purified from equal numbers of the control and apatinib treatment cells. Data are reported as the mean ± standard error (SD) from three independent experiments, *t*-test, ** *p* < 0.01, and *** *p* < 0.001.

**Figure 5 cancers-14-02492-f005:**
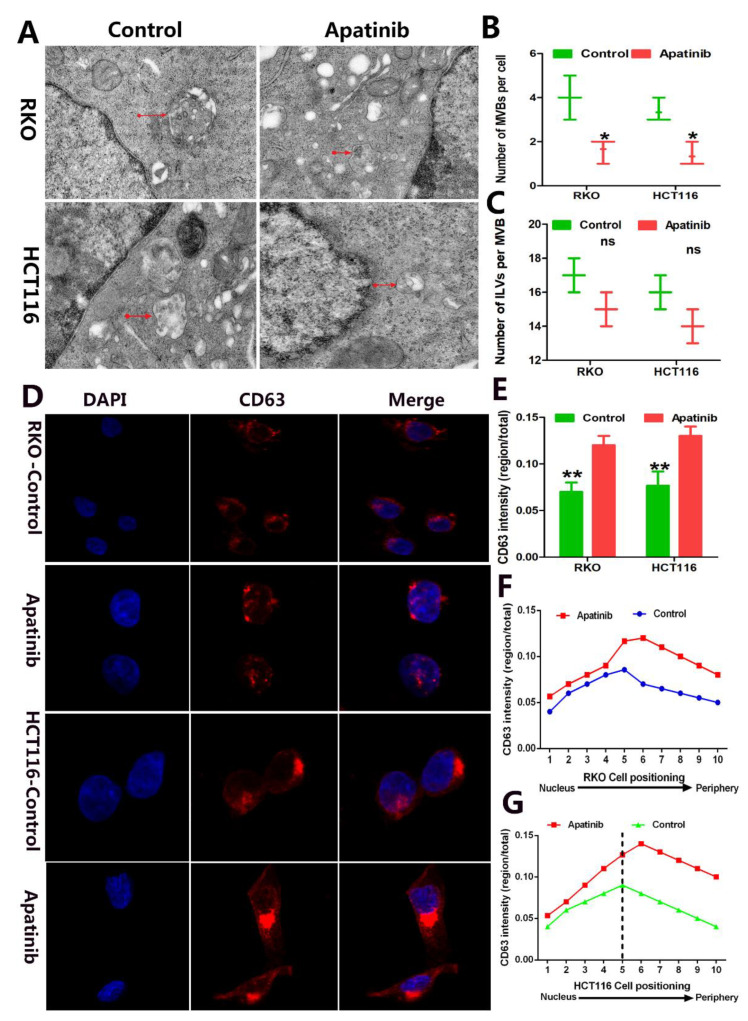
Apatinib inhibited the transport of MVBs towards the plasma membrane in CRC cells. (**A**) Representative electron microscopic images of control and apatinib treatment cells. (**B**) The number of MVBs per cell profile. (**C**) The number of ILVs per MVB. (**D**) Immunofluorescence of the control and apatinib-treated cells stained with CD63. (**E**) Quantification of CD63^+^ vesicle distribution within the cell. (**F**,**G**) Quantification of CD63 intensity within the perinuclear region of RKO and HCT116 cells. * *p* < 0.05, ** *p* < 0.01.

**Figure 6 cancers-14-02492-f006:**
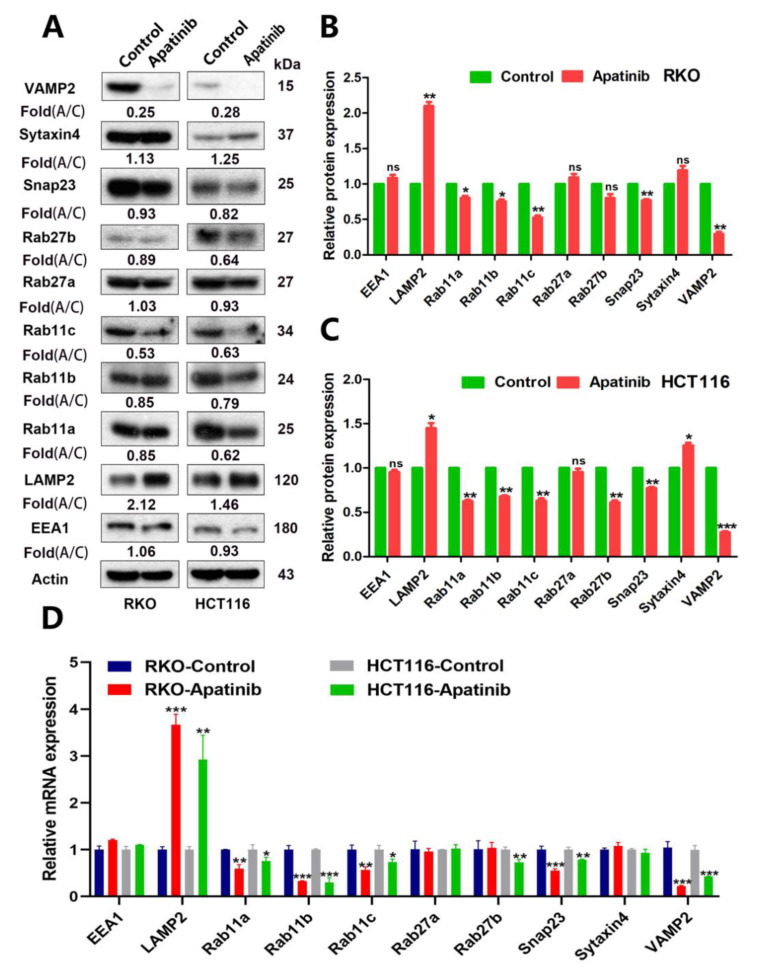
Apatinib regulated the expression of related molecules that were involved in the release of exosomes. (**A**–**C**) Western blotting analysis of exosome-releasing protein levels from the control and apatinib treatment cells. (**D**) Real-time PCR analysis of the mRNA expression of the exosome-releasing protein in the control and apatinib treatment cells. * *p* < 0.05, ** *p* < 0.01, and *** *p* < 0.001,ns: no significance.

**Figure 7 cancers-14-02492-f007:**
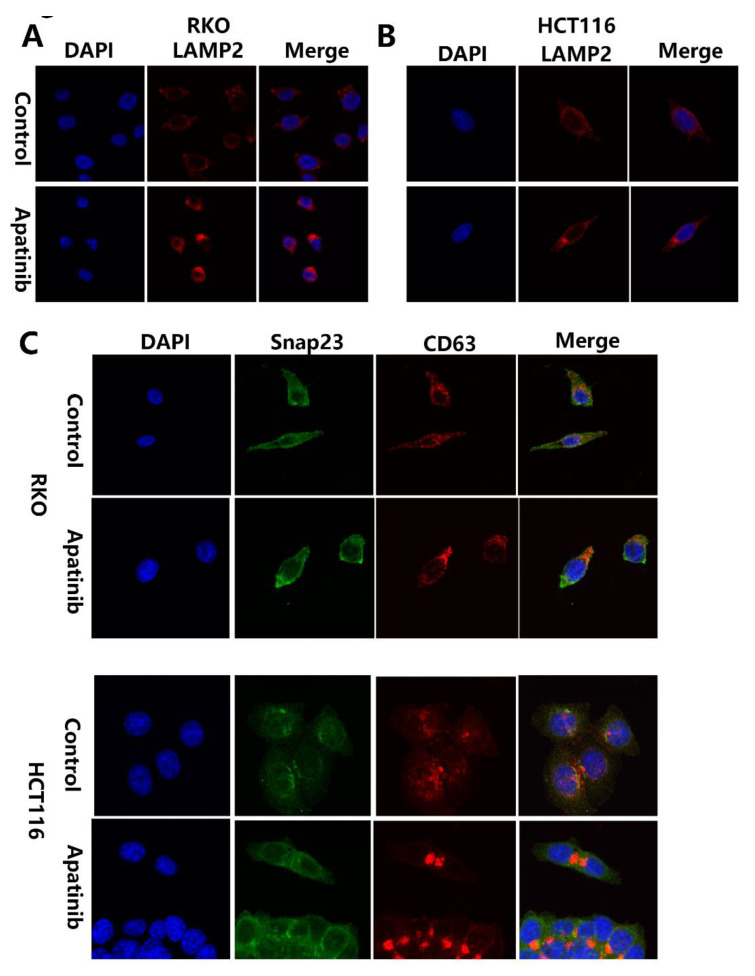
Apatinib regulates the localization of LAMP2 and colocalization of Snap23 with CD63. (**A**,**B**) Immunofluorescence of the LAMP2 (red) distribution in RKO and HCT116 cells with or without apatinib treatment. (**C**) Confocal colocalization analysis of CD63 (red) and SNP23 (green) in RKO and HCT116 cells with or without apatinib treatment.

**Figure 8 cancers-14-02492-f008:**
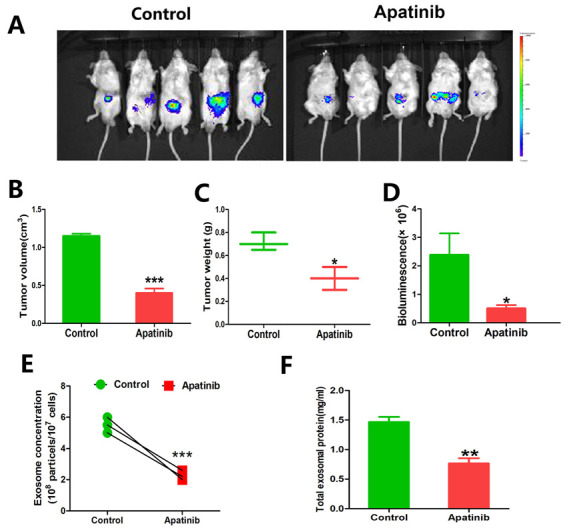
The antitumor efficacy of apatinib and exerted exosome inhibition in vivo. (**A**) Bioluminescent imaging for RKO-luc orthotopic xenograft colon tumors in the control vs. apatinib group (*n* = 5 per group). (**B**–**D**) Comparison of the tumor volume, tumor weight, and bioluminescence between the control and apatinib group. (**E**) NTA showed the concentrations of serum exosomes isolated from the control and apatinib group. (**F**) Quantification of the serum exosomal protein levels obtained from the control and apatinib treatment mice. * *p* < 0.05, ** *p* < 0.01, and *** *p* < 0.001.

**Table 1 cancers-14-02492-t001:** Patients baseline characteristics (*n* = 50).

Patient Characteristic	Value
Median age, years (range)	49 (37–68)
Sex	
Male	26 (52.0%)
Female	24 (48.0%)
ECOG performance status	
0–1	33 (66.0%)
2	17 (34.0%)
Primary cancer Site	
Left colon	26 (52.0%)
Right colon	12 (24.0%)
Rectum	12 (24.0%)
Histology	
Adenocarcinoma	48 (96.0%)
Mucinous carcinoma	1 (2.0%)
Undifferentiated carcinoma	1 (2.0%)
Metastatic Site	
Liver	33 (66.0%)
Lung	25 (50.0%)
Abdominal lymph node	8 (16.0%)
Peritoneum	16 (32.0%)
Other	15 (30.0%)
No. of metastatic sites	
≤2	17 (34.0%)
>2	33 (66.0%)
*KRAS* Status	
Wild type	23 (46.0%)
Mutated	27 (54.0%)
*BRAF* mutation	
No	48 (96.0%)
Yes	2 (4.0%)
Prior targeted treatments	
Neither	15 (30.0%)
Bevacizumab and Cetuximab	5 (10.0%)
Bevacizumab only	25 (50.0%)
Cetuximab only	5 (10.0%)

ECOG, Eastern Cooperative Oncology Group.

**Table 2 cancers-14-02492-t002:** Low-dose apatinib treatment-related adverse events.

Adverse Events	All Grades	Grade 3 to 4
Hypertension	16 (32%)	4 (8%)
Hand–foot skin reaction	15 (30%)	3 (6%)
Proteinuria	17 (34%)	1 (2%)
Dysphonia	6 (12%)	0
Diarrhea	3 (6%)	0
Mucositis oral	4 (8%)	0
Decreased appetite	10 (20%)	0
Platelet count decreased	2 (4%)	0
Neutropenia	2 (4%)	0
Anemia	3 (6%)	0
Fatigue	8 (16%)	0

## Data Availability

Data is contained within the article or Appendix A.

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
