# Peer review of "Studies of the Efficacy of Low-Dose Apatinib Monotherapy as Third-Line Treatment in Patients with Metastatic Colorectal Cancer and Apatinib’s Novel Anticancer Effect by Inhibiting Tumor-Derived Exosome Secretion"

_cancers, 2022, doi:10.3390/cancers14102492_

Round 1

Reviewer 1 Report

In this paper, the author found apatinib promoted degradations of MVBs via regulation of LAMP2, interfered with MVBs transport by inhibiting Rab11 expression. The author also showed apatinib inhibited MVBs membrane fusion by reducing SNAP23 and VAMP2 expression. The following comments may help to improve this paper.

The biggest concern of this paper is lacking a control group. In this study, all 50 patients were taken 250 mg apatinib orally once daily. As a comparison, the author should use the normal dosage of apatinib (425 mg to 750 mg once a day) as a positive control group. If the low dosage (250 mg once per day) is similar to the normal dosage (425 mg to 750 mg once a day) of apatinib, the author can draw the conclusion of low dose apatinib had comparable survival outcomes which the author declared in many places of this paper. 

Minor:
In Fig 2B, how does the author normalize the data, what is the scale bar of the heatmap indicated?

Reviewer 2 Report

Thank you for the opportunity to review this interesting article regarding apatinib, a promising VEGFR-2 inhibitor that potentially is a suitable candidate as a third-line treatment. 

The paper mainly consists of 2 different studies, one clinical that analyses the efficiency of the drug in low dose with less adverse events and a more solid translational study that tries to explain why the drug potentially has some benefits. 

Although I tried to find a link between the two studies, I believe they are totally independent. Therefore, I would suggest to the authors explain and/or correlate the clinical and translational parts of the study in order to have more linear writing. A reorganization of the chapters putting the clinical part before the conclusion can have some benefits. 

Regarding the clinical part, here are some of my concerns 

 1.       The study consists of 50 patients all taken 250 mg of apatinib. No control group with a different third-line treatment or/and patient with a higher or lower dosage of apatinib has been used to compare the result.   

2.       No median survival rate of these patients has been given. 

3.       Overall results hardly give any conclusion if patients had any benefit from this study. 

Regarding the translational part, the study, as mentioned, has some in vivo and in vitro results that potentially help to understand the mechanism of action. 

1.       I would suggest, if possible, to use a western blot with less exposure time in Fig 4F CD63 of EXO-HCT116 cells. 

2.       An explanation of why a unique dose and time (20 μM and 48 hours) has been used to show the inhibition of exosomes secretion would be useful. Did you use other doses of the drug? 

Overall in vitro and in vivo experiments are very interesting but the clinical conclusions can be improved or at least can be reorganized. 

Round 2

Reviewer 1 Report

No further questions.